# Relationships between Physical and Social Behavioural Changes and the Mental Status of Homebound Residents in Hong Kong during the COVID-19 Pandemic

**DOI:** 10.3390/ijerph17186653

**Published:** 2020-09-12

**Authors:** Ben Y. F. Fong, Martin C. S. Wong, Vincent T. S. Law, Man Fung Lo, Tommy K. C. Ng, Hilary H. L. Yee, Tiffany C. H. Leung, Percy W. T. Ho

**Affiliations:** 1School of Professional Education and Executive Development, The Hong Kong Polytechnic University, Hong Kong, China; ben.fong@cpce-polyu.edu.hk (B.Y.F.F.); vincent.law@cpce-polyu.edu.hk (V.T.S.L.); hilary.yee@speed-polyu.edu.hk (H.H.L.Y.); 2The Jockey Club School of Public Health and Primary Care, The Chinese University of Hong Kong, Hong Kong, China; wong_martin@cuhk.edu.hk; 3Department of Mathematics and Information Technology, The Education University of Hong Kong, Hong Kong, China; lomf@eduhk.hk; 4Faulty of Business, City University of Macau, Macau, China; tiffanyleung@cityu.mo; 5Department of Orthopaedics & Traumatology, The Chinese University of Hong Kong, Hong Kong, China; percyho@link.cuhk.edu.hk

**Keywords:** COVID-19, physical health changes, social contact, social distancing, Patient Health Questionnaire-9

## Abstract

In Hong Kong, social distancing has been adopted in order to minimise the spread of COVID-19. This study aims to examine the changes in physical health, mental health, and social well-being experienced by local residents who were homebound during the pandemic. An online questionnaire in both Chinese and English versions was completed by 590 eligible participants from 24 April to 13 May 2020. The questionnaire found that individuals aged 18 to 25 years spent more time resting and relaxing but experienced more physical strain. Working status was associated with social contact, with participants working full-time jobs scoring higher in “maintaining social communication via electronic means” and “avoiding social activities outside the home”. Additionally, approximately one third of the participants (29.7%) had moderate to severe depression, and participants aged 18 to 25 were found to have higher scores in PHQ-9. Changes in physical health and social contact were significantly associated with developing depressive symptoms. From the results, it is clear that the COVID-19 pandemic has the potential to exert a negative impact on the mental health status of individuals.

## 1. Introduction

The outbreak of Coronavirus Disease 2019 (COVID-19) was first reported in Wuhan, China, as a pneumonia of unknown cause. The World Health Organization (WHO) named the new virus COVID-19 on 11 February 2020 and declared the outbreak a world pandemic on 11 March 2020 [1]. COVID-19 causes respiratory infections with symptoms ranging from common cold-like to more severe diseases like Severe Acute Respiratory Syndrome (SARS) [2]. It is mainly transmitted through contact with respiratory droplets. However, many people only experience mild symptoms in the early stage, and the estimated incubation period ranges from 1 to 14 days [3]. By 31 August 2020, there were more than 25,000,000 confirmed cases in more than 200 countries, areas, or territories, with over 840,000 deaths reported.

In view of the rapid spread of COVID-19 around the world, experts have emphasised the importance of social distancing, which includes keeping at least one metre of distance between oneself and another person, especially among those who are ill. National ‘lockdowns’, quarantine measures, and curfews have been implemented in many countries, including Spain, Italy, Australia, and France, in order to minimise social human contact and avoid the spreading of the disease [4]. Schools, non-essential services, and factories have been shut down, and local residents are only allowed to go out for work, shopping, or medical appointments in circumstances which are considered necessary [5]. In Hong Kong, measures have included reducing the seating capacity at eateries, and temporarily closing high-risk premises such as bathhouses and fitness centres. Local residents have been encouraged to work from home, and in-person classes at all schools have been suspended [6].

To avoid the COVID-19 infection, residents may experience behavioural changes such as avoiding crowds, taking extra measures to maintain personal hygiene, and staying up to date with coronavirus news. The lives of the general public have been greatly disrupted and restricted in terms of their mobility, social interactions, and the performance of their daily activities. Many individuals have been suddenly “forced” to study or work from home and may experience changes in physical and social health. In terms of physical health, it has been revealed that adequate physical activity can prevent the onset of depression symptoms and stress [7,8]. Remaining homebound during social distancing periods can reduce physical activity and deteriorate physical fitness, but residents may in turn focus more on maintaining a healthy diet and a daily routine. The change in social relationships—for example the inability to support and connect with friends and family members—is also associated with depression [9,10]. Homebound residents are likely to experience negative changes in their social relationships due to reduced engagement in social activities and increased reliance on maintaining social contact via electronic means. In addition, working from home has been found to have mixed effects on psychological attitude and work-life balance, with some people benefitting from the integration of work and non-working life, and others feeling stressed due to the blurred boundaries between work and non-work [11,12].

Recent studies have shown that the COVID-19 pandemic has caused a substantial increase in depression, anxiety, and mild stress. Additionally, people with a higher education level experienced more stress [13,14,15,16,17]. The uncertainty around the COVID-19 outbreak and the sudden change in studying and working environments could lead to the development of negative emotions such as anxiety, which can adversely influence individual mental health status. However, family support and the use of social networking sites are factors associated with the level of depression and anxiety; therefore, whether residents are receiving sufficient social support and are in a state of positive connectedness could affect their mental health status during the pandemic [9,17,18]. Although there is ongoing clinical research on the symptoms of patients infected by COVID-19 and its incubation period and transmission mode [19,20,21], limited studies have focused on the impact of the pandemic on changes in people’s behaviour and mental status—especially those who are homebound. Therefore, this study aims to examine the changes and relationships between physical health, mental health, and social well-being of the general public who are homebound due to arrangements of working and studying from home during the COVID-19 pandemic.

## 2. Materials and Methods 

### 2.1. Participants and Data Collection

Participants were recruited exclusively through online methods using Google Forms. The questionnaire link was sent and shared amongst staff and students of Hong Kong Community College and the School of Professional Education and Executive Development at The Hong Kong Polytechnic University. The questionnaire was sent via email and a WhatsApp group used to share health-related information with informed consent. Only those who were Hong Kong residents and had been homebound for at least four days a week due to COVID-19 at the time of study were recruited. Those who are cognitively impaired were excluded in this study. Data was collected from 24 April to 13 May 2020.

### 2.2. Survey Instrument

A questionnaire comprising four parts, in both English and Chinese versions, was posted online. The first part collected demographic data, covering age, gender, average daily studying or working hours at home, and changes in daily routines. The second part—“physical health changes”—consisted of five questions about changes in physical health, rated in terms of “Much decreased”, “Decreased”, “No change”, “Increased’, and “Much increased”. The third part was comprised of seven items about the aspects of social contact and social support. The fourth part on mental status was made up of 10 items, nine of which were adopted from the validated Patient Health Questionnaire-9 (PHQ-9). The additional question enquired about the increase of stress due to working from home. Participants were required to answer nine questions related to the frequency of depressive symptoms by selecting one of the following options: “Not at all”, “Several days”, “More than half the days” or “Nearly every day” [22]. The severity of depressive symptoms was determined by the total score ranging from 0 to 27. A higher score represents a greater level of depressive symptoms. Participants were classified into five different groups based on the severity of depressive symptoms: (i) having minimal or no depressive symptoms (0–4); (ii) mild (5–9); (iii) moderate (10–14); (iv) moderately severe (15–19); and (v) severe depressive symptoms (20–27). Typically, participants with scores of 10 and above would be advised to seek medical attention.

### 2.3. Statistical Analysis

Descriptive statistics for demographic variables including household members, education level, working status, and changes in activities of daily living were reported in frequencies and proportions. Statistical analysis was performed using Statistical Package for the Social Sciences (SPSS) Statistics (Version 25, IBM Crop., Armonk, NY, USA).

Chi-square tests of independence were used to test for statistical significance between categorical variables. An independent *t*-test was used to determine whether there was a difference in gender and score regarding physical, social, and mental aspects. One-way Analysis of Variance (ANOVA) was employed to assess the difference in working status and total score on physical, social, and mental status. Pearson’s correlation coefficient was used to investigate the relationship between social change, physical change, and mental status. Multivariate regression models were used to evaluate the covariates significantly associated with the four outcome variables (i.e., changes in physical health, social contact, social support and depressive severity) separately. Ten dummy demographic variables were coded to be included in the regression model. All *p*-values < 0.01 were considered to be statistically significant. To evaluate the reliability of this study, the internal consistency was examined and the Cronbach’s alpha values of the multi-item scales were considered acceptable when the value was at least 0.5 [23].

## 3. Results

### 3.1. Participant Characteristics

Seven hundred and eighty seven (787) responses were received, with 590 eligible participants in total. The demographic factors are presented in Table 1. A plurality of participants were 18 to 25 years old (46.9%) and had attained a tertiary education level or above (86.5%). 95.9% of participants stayed at home more often and 71.7% paid more attention to personal hygiene.

### 3.2. Physical Health Changes

The Cronbach’s alpha of this construct was 0.554, which was above the cut-off value of 0.5 [24]. A t-test of independence showed no significant differences between gender and total score in physical health change (*t* = 0.358, df = 588, *p* = 0.720), but individuals who were studying scored higher in physical health changes than those who were not studying (*t* = 2.727, df = 588, *p* < 0.05). However, chi-square tests showed a significant difference between age group and “time to rest”, χ^2^ (24, *n* = 590) = 48.713, *p* = 0.002 (Table 2). 40.2% of participants experienced an increase in the amount of time they spent relaxing, while 51.0% spent more time resting. Overall, individuals aged 18 to 25 spent much more time relaxing and resting as reflected by the examination of the observed and expected frequency count (f_o_ = 33 and f_e_ = 19.7; f_o_ = 125 and f_e_ = 112.7). This group also experienced an increase in physical strain (f_o_ = 109 and f_e_ = 87.8).

In multivariate regression models, three variables were significantly associated with a change in physical health (Table 3). The most important factor in predicting a positive change in physical health was being younger than 18 years old with a coefficient of 1.435, which was significantly higher than the other age groups.

### 3.3. Social Health Changes

The Cronbach’s alpha of items regarding social contact and social support were 0.507 and 0.785 respectively, both of which were above the recommended value [24]. There was a significant difference between gender and total score in social contact, but the effect was small, with females (*M* = 6.02) scoring higher than males (*M* = 5.60), (*t* = 3.147, df = 588, *p* < 0.005, *d* = 0.3). Since the data was not normally distributed and the variance was heterogeneous, a Robust Welch ANOVA was conducted to examine the effect of participants’ working status on their total score in “maintaining social communication via electronic means” and “avoiding social activities outside the home”. This showed a significant effect of presentation condition: *F*(2235.1) = 33.42, *p <* 0.001. From post-hoc Games-Howell analysis, the mean of social health changes in participants who were studying was significantly higher than that of the working and underemployed groups. 

There was a significant relationship between age group and “maintain social communication via electronic means”, χ^2^ (24, *n* = 590) = 90.751, *p <* 0.001, “avoid social activities outside the home”: χ^2^ (24, *n* = 590) = 97.611, *p* = 0.000, “share feelings with family members”: χ^2^ (24, *n* = 590) = 47.801, *p* = 0.003, and “care for family members’ feelings”: χ^2^ (24, *n* = 590) = 53.474, *p* = 0.001 (Table 4). 

Nearly half of the participants increased their usage of electronic tools to communicate with others (48.6%), while just over half increased their avoidance of social activities outside their homes (50.8%). However, 45.8% of participants experienced no change in sharing feelings with family members or addressing their emotional health. Overall, participants aged 18 to 25 years increased their use of electronic means to maintain communication (f_o_ = 119 and f_e_ = 82.6) and increased their avoidance of social activities (f_o_ = 154 and f_e_ = 140.8) when compared with other age groups. 

Individuals aged below 55, females, and degree- and higher degree-holders were found to be positively and significantly associated with an increase in the usage of electronic tools and the avoidance of outside activities. However, the association between demographic variables and social support was not significant. 

### 3.4. Mental Health Status

The Cronbach’s alpha of ten items was 0.926, which demonstrated good internal consistency [24]. The assumptions of normality and homogeneity of variance were violated and therefore a Robust Welch ANOVA was conducted to examine the effect of working status on total PHQ-9 score. This revealed a significant effect of presentation condition: *F*(2587) = 51.934, *p* < 0.001. By comparing means in post-hoc Games-Howell, individuals who were studying scored higher than those in the working and underemployed groups.

A chi-square test showed that there was a significant relationship between age group and depressive severity: χ^2^ (4, *n* = 590) = 123.801, *p* < 0.001, as well as between educational level and depressive severity: χ^2^ (4, *n* = 590) = 25.904, *p* < 0.001 (Table 5). 65.3% of participants experienced increased stress due to staying at home and 29.7% experienced moderate to severe levels of depressive symptoms. Overall, a significantly higher number of participants aged 18 to 25 years old than expected had moderate (f_o_ = 61 and f_e_ = 39.9) and moderately severe (f_o_ = 45 and f_e_ = 26.8) depressive symptoms. Most participants in the 56 to 65 age group showed minimal or no depressive symptoms based on the examination of the observed and expected frequency count (f_o_ = 61 and f_e_ = 37.5). 

From the result of multivariate regression, social contact was found to be significantly associated with the total score of PHQ-9, indicating that the increase in the use of electronic devices and the decrease in outside activities was positively associated with a higher level of depressive severity. However, physical health changes were negatively associated with depressive severity, meaning that individuals with more time to relax and rest were at a lower risk of experiencing depressive symptoms. The 18 to 25 age group scored significantly higher in the total score of PHQ-9 than the below-18 age group.

### 3.5. Relationship between Changes in Physical and Social Health and Mental Health Status

The relationship between changes in physical and social health, and mental health status is shown in Table 6. There was a significant positive correlation between total physical score and social contact score (r = 0.146, *n* = 590, *p* < 0.001), and total physical score and social support score (r = 0.194, *n* = 590, *p* < 0.001). Total social contact score was also positively correlated with total PHQ-9 score (r = 0.211, *n* = 590, *p* < 0.001) and social support score (r = 0.220, *n* = 590, *p* < 0.001). Also, there was a significant negative correlation between the total physical score and the PHQ-9 score (r = −0.097, *n* = 590, *p* < 0.05).

## 4. Discussion

Since COVID-19 became a pandemic, people have been urged to maintain social distance and increase personal and environmental hygienic habits to mitigate the transmission of the coronavirus. These measures enable individuals to avoid the inhalation of droplets expelled by infected persons as well as contact with contaminated surfaces, objects, and body parts [25]. The Hong Kong Government has implemented special work and online teaching arrangements for government departments and schools respectively to reduce social contact and, thus, the risk of spreading the novel coronavirus in the community. Employees are allowed to work from home, and all face-to-face teaching in educational institutions has been suspended. This study indicated that 95.9% of participants stayed at home more often to maintain social distancing, and over 70% paid more attention to their personal hygiene during the pandemic, which is similar to the health behaviour changes noted during the outbreak of SARS in Hong Kong in 2003 [25]. After experiencing the SARS epidemic, over 98% and 70% of the participants in Hong Kong reported the adoption of good personal hygiene and the wearing of face masks, respectively [26]. This practice has likely influenced the everyday routines of most Hong Kong residents, and in turn resulted in good personal hygiene maintenance during the current pandemic. However, this study showed that being homebound did not change participants’ behaviours regarding diet or physical exercise. It is possible that they believed they were already in tune with their health, or that it takes a longer period of time to change diet and exercise habits. Additionally, participants who studied or worked from home needed to rely more on electronic devices to fulfil their work commitments or attend classes, resulting in additional use of the Internet and digital screens.

This study found that 29.7% of participants reported moderate to severe depressive symptoms and this result was similar to studies conducted in China and Italy, which found that 25% and 32.8% of the participants, respectively had moderate to severe depressive symptoms during the COVID-19 pandemic [27,28]. The level of depressive symptoms can be affected by the risk perception of health crisis. Risk perception is subjective judgement of possible consequences of an event, influenced by cognitive, emotional, social and cultural variation between different individuals and countries [29,30]. Personal knowledge, trust in government, healthcare professionals and science, and information of COVID-19 collected from friends and family are significant predictors of risk perception [30]. Additionally, affective risk perception, which is strongly subjective, is positively associated with the depressive symptoms during the COVID-19 pandemic, while negative association is found between cognitive risk perception, meaning individual judgement is based on scientific support, and depressive symptoms [27]. Risk perception may result in changes in daily routines such as paying more attention to personal hygiene and browsing more prevention information during the pandemic, hence, affecting depressive symptoms. 

Owing to school and university closures, students aged between 18 to 25 stayed at home to study and attended online lessons and courses. Unsurprisingly, this study found that younger participants tended to use electronic devices more often to maintain their social life without physical contact with classmates and friends. They used electronic devices to communicate with their friends and lecturers and spent more time using online learning platforms [31]. Such learning modes facilitated the interaction between students and lecturers in receiving more feedback related to their assignments and course progress [32]. Moreover, younger people tended to have higher usage rates of electronic products than older people. They were more likely to visit different social media platforms [33,34] and, therefore, may have preferred staying at home and relying on electronic devices to obtain information and seek entertainment. This group also experienced increased physical strain, which could be explained by their more frequent usage of electronic devices; often a cause of prolonged physical discomfort such as neck, shoulder, and upper extremity pain [35]. 

This study also found a significant difference in social contact changes according to gender. Females tended to have higher scores in social contact, meaning that they communicated with friends and family members by using electronic devices and avoided outdoor social activities. This result is in line with that of another study which found that females were more frequent users of social media, texting, and online video calls for social connectivity [36]. This may be due to the gender differences in displaying emotions, as females tend to express their emotions of happiness, sadness, and fear to others [37]. In view of the COVID-19 pandemic, social networking and online communication tools, such as Facebook and WhatsApp, could be useful platforms to enhance communication and social connection among friends and family members.

Participants demonstrated no significant increase in sharing feelings and caring for family members. This is not in line with previous studies which showed that people shared feelings more often with family and friends during a health disease crisis [13,38]. In particular, individuals aged 18 to 25 indicated no change in such social support. People might avoid discussing COVID-19 to minimise the trigger of negative emotions such as anxiety regarding self-protection [39]. It may also be due to easy access to health information and news, and hence people feel unnecessary to share related feelings with others.

More importantly, the 18 to 25 age group felt increased levels of stress due to studying at home for more than half of each day, and almost every day. Although the participants who were studying, and thus using electronic devices, were maintaining social contact, they still showed moderate and moderately severe levels of depressive symptoms. A similar trend among young people was found by another local study which discovered that people who had not experienced the SARS outbreak in 2003 were associated with a poorer mental health status [11]. Scores of participants who were studying were higher than those who were working and underemployed. These participants were also predicted to have higher scores in PHQ-9 than other age groups. In contrast, participants aged 56 or above indicated no increase of stress arising from working from home. This may be because those aged 18 to 25 years had not experienced such a learning arrangement before and were more likely to become worried about their academic progress, learning support, and the teaching materials [15,40], as well as about when normal classes would resume. Younger people also tend to use social media more, which might trigger more stress. Further support provided by educational institutions, teachers, and peers may help to optimize students’ learning progress, and thus relieve the stress and depressive symptoms induced by studying at home [40]. Using telemedicine to provide psychological counselling can also be enhanced to reduce psychological distress [14]. 

Those aged 56 and above experienced much lower levels of depressive symptoms, which could indicate that they prefer working from home instead of going about their typical working day. In addition, this study could not find a significant difference between participants who were working and underemployed with respect to changes in mental health. This finding contrasts with another study which found that staying at home due to the COVID-19 pandemic was associated with greater financial concern and health anxiety [41]. Considering the uncertain economic environment caused by the pandemic, it would be expected that working individuals and underemployed participants would be more depressed and have higher anxiety levels than those who were studying. One possible reason for this unexpected outcome is that the participants had relatively stable jobs or were approaching retirement, therefore they showed less significant concern.

A negative correlation between physical health change and total PHQ-9 score was found in this study. This is similar to another study which showed that participants who made positive changes to their physical health, including increasing physical activity and maintaining healthier diets, were associated with not only better physical health but also better mental health [42]. An increase in physical activity was associated with a decrease in depressive symptoms which implied that physical activity at home could potentially relieve stress and depressive symptoms [43]. Furthermore, a positive association and correlation between social contact and total PHQ-9 score was found in this study, meaning that participants relying more on social contact via online platforms and fewer outdoor social activities might have higher levels of depressive symptoms. In fact, participants who had inadequate physical social interaction and a reduction of face-to-face communication during the pandemic might be more prone to depressive symptoms. Spending more time using screens was significantly associated with developing depressive symptoms [44]. The reduction of time spent having social interactions was correlated to the depressive symptoms [45], whilst infrequent social interaction could also predict higher depressive symptoms [46]. Therefore, participants who were homebound during the pandemic with insufficient social interactions might be more likely to develop depressive symptoms. 

This study has a number of limitations. First, the behavioural changes and mental status of participants were only measured during the pandemic, as the situation before the pandemic could not be identified. Future studies could aim to investigate and compare the difference of behavioural changes and mental status of participants after the pandemic. Second, socioeconomic variables including income and social class could be important variables through which the findings of this study could be interpreted. Third, 46.9% of the participants were aged 18 to 25 indicating selection biases. This could be due to the fact that this study was only accessible via an online platform; therefore, younger subjects were easier to recruit. The overrepresentation of younger subjects in this study is not necessarily representative of all residents in Hong Kong.

## 5. Conclusions

This is the first time that Hong Kong residents have experienced such quarantine and social distancing arrangements for more than two months. To our knowledge, this is the first study to examine the physical, social, and mental health changes of Hong Kong residents who stayed at home to study or work during the COVID-19 pandemic. Local residents experienced disruptions to their daily routines and changes in behaviour such as an increasing amount of time spent relaxing and resting. Stress level among younger age (18–25) people are higher than that of the other age bands due to worry about academic progress and learning support. Likewise, more than three quarters of the participants chose electronic means to maintain their social contact as well as avoiding outdoor social activities. It is noteworthy that nearly a third of the participants in this study experienced moderate to severe depressive symptoms. Such high prevalence of depressive symptoms should provide the Government and health professionals with some insight into the mental status of the local residents, particularly students. Further studies to investigate depression among Hong Kong students during and after major social events like the pandemic are warranted. Moreover, the findings of this study have implications on practices in community health and in the Government’s preparedness and response plan for the outbreak [47], particularly in the early identification and management of depression among at-risk individuals. 

## Figures and Tables

**Table 1 ijerph-17-06653-t001:** Characteristics of study participants.

Characteristic	Frequency	Percentage
*Gender*		
Male	217	36.8%
Female	373	63.2%
*Age*		
Below 18	15	2.5%
18–25	277	46.9%
26–35	53	9.0%
36–45	62	10.5%
46–55	63	10.7%
56–65	78	13.2%
66 or above	42	7.1%
*No. of Household Members*		
0	10	1.7%
1	40	6.8%
2–3	268	45.4%
4–5	242	41.0%
6 or above	30	5.1%
*Education Level*		
Primary or less	2	0.3%
Secondary	78	13.2%
Tertiary	258	43.7%
Degree	147	24.9%
Postgraduate or above	105	17.8%
*Working status*		
Studying	281	47.6%
Working	216	36.6%
Underemployed	93	15.8%
*Changes in daily routines during COVID-19 pandemic*		
Staying more at home	566	95.9%
Paying more attention to personal hygiene (i.e., washing hands, wearing mask and using alcohol wipes)	423	71.7%
Spending more time on screen (e.g., computers, electronic devices, etc.) and internet	404	68.5%
Avoiding crowds and social gathering	382	64.7%
Browsing COVID-19 news and prevention information	347	58.8%
Engaging in social distancing	343	58.1%
Performing less physical activities than usual	277	46.9%
Preparing food by yourself more frequent than usual	248	42.0%
Searching more information of buying surgical masks	245	41.5%
Browsing more health-related news	195	33.1%
More adherence to experts and government recommendations	147	24.9%
Praying more often at home	63	10.7%

**Table 2 ijerph-17-06653-t002:** Relationship of physical health changes by demographic factors.

Variables	Age Group (Years) (*n* = 590)								Educational Level (*n* = 590)		
	Below 18 (*n* = 15)	18–25 (*n* = 277)	26–35 (*n* = 53)	36–45 (*n* = 62)	46–55 (*n* = 63)	56–65 (*n* = 78)	66 or Above (*n* = 42)	*p*-Value ^1^	Post-Secondary or Below (*n* = 338)	Degree or Above (*n* = 252)	*p*-Value ^1^
Time spent to relax e.g., doing leisure activities								0.037			0.788
*Much decreased*	0 (−1.0)	19 (−0.2)	2 (−1.7)	8 (+3.7)	5 (+0.6)	4 (−1.4)	3 (+0.1)		25 (+1.5)	16 (−1.5)	
*Decreased*	2 (−0.4)	47 (+2.4)	8 (−0.5)	6 (−4.0)	10 (−0.1)	13 (+0.4)	9 (+2.2)		57 (+2.6)	38 (−2.6)	
*No change*	5 (−0.5)	87 (−14.9)	18 (−1.5)	25 (+2.2)	26 (+2.8)	36 (+7.3)	20 (+4.6)		117 (−7.3)	100 (+7.3)	
*Increased*	6 (+1.0)	91 (−0.6)	23 (+5.5)	20 (−0.5)	21 (+0.2)	24 (−1.8)	10 (−3.9)		114 (+2.3)	81 (−2.3)	
*Much increased*	2 (+0.9)	33 (+13.3)	2 (−1.8)	3 (−1.4)	1 (−3.5	1 (−4.6)	0 (−3.0)		25 (+0.9)	17 (−0.9)	
Time spent to rest e.g., taking naps and sleeping								0.002			0.653
*Much decreased*	0 (−0.3)	6 (+0.4)	0 (−1.1)	4 (+2.7)	2 (+0.7)	0 (−1.6)	0 (−0.9)		5 (-1.9)	7 (+1.9)	
*Decreased*	0 (−1.0)	20 (+1.7)	4 (+0.5)	3 (−1.1)	7 (+2.8)	2 (−3.2)	3 (+0.2)		23 (+0.7)	16 (−0.7)	
*No change*	7 (+0.9)	88 (−23.7)	23 (+1.6)	23 (−2.0)	29 (+3.6)	**45 (+13.5)**	23 (+6.1)		131 (−5.3)	107 (+5.3)	
*Increased*	4 (−2.1)	**125 (+12.3)**	20 (−1.6)	28 (+2.8)	22 (−3.6)	25 (−6.7)	16 (−1.1)		144 (+6.5)	96 (−6.5)	
*Much increased*	4 (+2.4)	38 (+9.4)	6 (+0.5)	4 (−2.4)	3 (−3.5)	6 (−2.1)	0 (−4.3)		35 (+0.1)	26 (−0.1)	

The differences between obtained values and that expected with the Chi-square test of independence are represented in brackets. Data in bold are significantly overrepresented in comparison to predicted values (*p* < 0.01); ^1^ Chi-square test.

**Table 3 ijerph-17-06653-t003:** Regression model for predicting physical health changes, social contact, social support and depressive severity.

Outcomes	Unstandardised Coefficients		*t*	Sig.	R^2^	Adjusted R^2^
	B	Standard Error				
Physical health changes				0.040	0.032	0.015
(Constant)	4.250	0.265	16.069	0.000		
Age_dummy_1	1.435	0.547	2.623 **	0.009		
Age_dummy_2	0.932	0.373	2.496 *	0.013		
Age_dummy_3	0.724	0.351	2.063 *	0.040		
Age_dummy_4	0.384	0.346	1.110	0.267		
Age_dummy_5	0.231	0.334	0.690	0.490		
Age_dummy_6	0.362	0.298	1.215	0.225		
Gender_dummy	−0.075	0.133	−0.559	0.576		
Edu_dummy	0.102	0.151	0.673	0.501		
Work_dummy_1	−0.297	0.335	−0.887	0.376		
Work_dummy_2	−0.212	0.230	−0.919	0.358		
Social contact				0.000	0.186	0.172
(Constant)	4.097	0.246	16.640	0.000		
Age_dummy_1	1.719	0.509	3.377 **	0.001		
Age_dummy_2	1.775	0.348	5.107 **	0.000		
Age_dummy_3	1.479	0.327	4.528 **	0.000		
Age_dummy_4	0.980	0.322	3.042 **	0.002		
Age_dummy_5	0.953	0.311	3.066 **	0.002		
Age_dummy_6	0.458	0.277	1.652	0.099		
Gender_dummy	0.357	0.124	2.872 **	0.004		
Edu_dummy	0.455	0.141	3.230 **	0.001		
Work_dummy_1	0.189	0.312	0.606	0.545		
Work_dummy_2	−0.048	0.214	−0.222	0.824		
Social support				0.454	0.017	0.000
(Constant)	11.077	0.429	25.797	0.000		
Age_dummy_1	0.845	0.888	0.951	0.342		
Age_dummy_2	−0.117	0.606	−0.193	0.847		
Age_dummy_3	−0.181	0.570	−0.318	0.750		
Age_dummy_4	−0.090	0.562	−0.161	0.872		
Age_dummy_5	−0.014	0.542	−0.025	0.980		
Age_dummy_6	0.483	0.483	0.999	0.318		
Gender_dummy	0.432	0.217	1.995	0.047		
Edu_dummy	0.319	0.246	1.296	0.195		
Work_dummy_1	0.362	0.544	0.665	0.506		
Work_dummy_2	0.280	0.374	0.750	0.453		
Depressive severity				0.000	0.206	0.201
(Constant)	4.981	1.129	4.411	0.000		
Age_dummy_1	3.284	1.581	2.078 *	0.038		
Age_dummy_2	5.425	0.525	10.338 **	0.000		
Age_dummy_3	0.054	1.382	0.057	0.873		
Age_dummy_4	0.018	0.471	0.019	0.882		
Age_dummy_5	−0.001	−0.033	−0.001	0.880		
Age_dummy_6	−0.045	−1.126	−0.047	0.840		
Gender_dummy	−0.005	−0.127	−0.005	0.981		
Edu_dummy	−0.027	−0.667	−0.028	0.810		
Work_dummy_1	0.090	1.082	0.045	0.197		
Work_dummy_2	−0.040	−0.834	−0.035	0.592		
Physical Health Changes	−0.705	0.160	−4.404 **	0.000		
Social Contact	0.441	0.165	2.671 **	0.008		
Social Support	0.038	0.989	0.041	0.910		

Age_dummy_1: Dummy for below 18 age group; Age_dummy_2: Dummy for 18–25 age group; Age_dummy_3: Dummy for 26–35 age group; Age_dummy_4: Dummy for 36–45 age group; Age_dummy_5: Dummy for 46–55 age group; Age_dummy_6: Dummy for 56–65 age group; Gender_dummy: Dummy for female respondents; Edu_dummy: Dummy for degree or above respondents; Work_dummy_1: Dummy for studying respondents; Work_dummy_2: Dummy for working respondents; Note. * *p* < 0.05, ** *p* < 0.01.

**Table 4 ijerph-17-06653-t004:** Relationship of social health changes by demographic factors.

Variables	Age Group (*n* = 590)								Educational Level (*n* = 590)		
	Below 18 (*n* =15)	18–25 (*n* = 277)	26–35 (*n* = 53)	36–45 (*n* = 62)	46–55 (*n* = 63)	56–65 (*n* = 78)	66 or Above (*n* = 42)	*p*-Value ^1^	Post-Secondary or Below (*n* = 338)	Degree or Above (*n* = 252)	*p*-Value ^1^
Social contact											
Maintain social communication via electronic means e.g., telephone, e-mail, and social media								0.000			0.232
*Much decreased*	0 (0.0)	0 (−0.5)	1 (+0.9)	0 (−0.1)	0 (−0.1)	0 (−0.1)	0 (−0.1)		0 (−0.6)	1 (+0.6)	
*Decreased*	0 (−0.3)	4 (−1.6)	0 (−1.1)	3 (+1.7)	1 (−0.3)	2 (+0.4)	2 (+1.1)		6 (−0.9)	6 (+0.9)	
*No change*	2 (−0.9)	31 (−22.5)	7 (−3.2)	11 (−1.0)	19 (+6.8)	**28 (+12.9)**	16 (+7.9)		61 (−4.3)	53 (+4.3)	
*Increased*	8 (+0.7)	123 (−11.7)	30 (+4.2)	37 (+7.2)	29 (−1.6)	37 (−0.9)	23 (+2.6)		159 (−5.4)	128 (+5.4)	
*Much increased*	5 (+0.5)	**119 (+36.4)**	15 (−0.8)	11 (−7.5)	14 (−4.8)	11 (−12.3)	1 (−11.5)		112 (+11.2)	64 (−11.2)	
Avoid social activities outside the home								0.000			0.321
*Much decreased*	0 (−1.0)	2 (−16.3)	1 (−2.5)	7 (+2.9)	3 (−1.2)	**17 (+11.8)**	9 (+6.2)		23 (+0.7)	16 (−0.7)	
*Decreased*	1 (−0.6)	14 (−11.8)	3 (−1.9)	7 (+1.2)	11 (+5.1)	10 (+2.7)	9 (+5.1)		38 (+6.5)	17 (−6.5)	
*No change*	2 (+0.9)	23 (+3.3)	4 (+0.2)	2 (−2.4)	4 (−0.5)	4 (−1.6)	3 (0.0)		26 (+1.9)	16 (−1.9)	
*Increased*	7 (−0.6)	**154 (+13.2)**	29 (+2.1)	28 (−3.5)	33 (+1.0)	31 (−8.7)	18 (−3.4)		169 (−2.9)	131 (+2.9)	
*Much increased*	5 (+1.1)	**84 (+11.7)**	16 (+2.2)	18 (+1.8)	12 (−4.4)	16 (−4.4)	3 (−8.0)		82 (−6.2)	72 (+6.2)	
Social support											
Receive support from family members								0.119			0.057
*Much decreased*	1 (+0.8)	6 (+2.7)	0 (−0.6)	0 (−0.7)	0 (−0.7)	0 (−0.9)	0 (−0.5)		6 (+2.0)	1 (−2.0)	
*Decreased*	0 (−0.4)	7 (−0.5)	2 (+0.6)	1 (−0.7)	3 (+1.3)	1 (−1.1)	2 (+0.9)		11 (+1.8)	5 (−1.8)	
*No change*	9 (−0.5)	165 (−10.6)	34 (+0.4)	36 (−3.3)	38 (−1.9)	60 (+10.6)	32 (+5.4)		217 (+2.7)	157 (−2.7)	
*Increased*	3 (−1.3)	83 (+3.7)	15 (−0.2)	23 (+5.2)	20 (+2.0)	17 (−5.3)	8 (−4.0)		86 (−10.8)	83 (+10.8)	
*Much increased*	2 (+1.4)	16 (+4.7)	2 (−0.2)	2 (−0.5)	2 (−0.6)	0 (−3.2)	0 (−1.7)		18 (+4.3)	6 (−4.3)	
Receive support from friends								0.451			0.757
*Much decreased*	0 (−0.2)	8 (+3.8)	0 (−0.8)	1 (+0.1)	0 (−1.0)	0 (−1.2)	0 (−0.6)		7 (+1.8)	2 (−1.8)	
*Decreased*	0 (−1.1)	22 (+1.3)	4 (0.0)	6 (+1.4)	8 (+3.3)	2 (−3.8)	2 (−1.1)		25 (−0.2)	19 (+0.2)	
*No change*	11 (+1.7)	161 (−11.3)	32 (−1.0)	39 (+0.4)	35 (+4.2)	58 (+9.5)	31 (+4.9)		212 (+1.8)	155 (−1.8)	
*Increased*	3 (−0.9)	77 (+4.2)	15 (+1.1)	15 (−1.3)	19 (+2.4)	17 (−3.5)	9 (−2.0)		86 (−2.8)	69 (+2.8)	
*Much increased*	1 (+0.6)	9 (+2.0)	2 (+0.7)	1 (−0.6)	1 (−0.6)	1 (−1.0)	0 (−1.1)		8 (−0.6)	7 (+0.6)	
Share feelings with family members								0.003			0.416
*Much decreased*	0 (−0.1)	3 (+0.7)	1 (+0.6)	0 (−0.5)	1 (+0.5)	0 (−0.7)	0 (−0.4)		3 (+0.1)	2 (−0.1)	
*Decreased*	0 (−0.5)	5 (−4.4)	5 (+3.2)	4 (+1.9)	4 (+1.9)	2 (−0.6)	0 (−1.4)		9 (−2.5)	11 (+2.5)	
*No change*	8 (−1.1)	**195 (+27.4)**	31 (−1.1)	30 (−7.5)	29 (−9.1)	40 (−7.2)	24 (−1.4)		215 (+10.5)	142 (−10.5)	
*Increased*	5 (+0.3)	63 (−24.3)	15 (−1.7)	25 (+5.5)	26 (+6.1)	34 (+9.4)	18 (+4.8)		100 (−6.6)	86 (+6.6)	
*Much increased*	2 (+1.4)	11 (+0.7)	1 (−1.0)	3 (+0.7)	3 (+0.7)	2 (−0.9)	0 (−1.6)		11 (−1.6)	11 (+1.6)	
Share feelings with friends								0.371			0.900
*Much decreased*	0 (−0.3)	7 (+2.3)	1 (+0.1)	1 (−0.1)	1 (−0.1)	0 (−1.3)	0 (−0.7)		7 (+1.3)	3 (−1.3)	
*Decreased*	0 (−1.1)	16 (−5.1)	6 (+2.0)	8 (+3.3)	8 (+3.2)	2 (−3.9)	5 (+1.8)		26 (+0.2)	19 (−0.2)	
*No change*	9 (+1.4)	139 (−0.4)	26 (−0.7)	32 (+0.8)	34 (+2.3)	36 (−3.3)	21 (−0.1)		166 (−4.1)	131 (+4.1)	
*Increased*	4 (−1.3)	98 (+0.3)	17 (−1.7)	20 (−1.9)	17 (−5.2)	37 (+9.5)	15 (+0.2)		122 (+2.8)	86 (−2.8)	
*Much increased*	2 (+1.2)	17 (+2.9)	3 (+0.3)	1 (−2.2)	3 (−0.2)	3 (−1.0)	1 (−1.1)		17 (−0.2)	13 (+0.2)	
Care for family members’ feelings								0.001			0.208
*Much decreased*	0 (−0.1)	1 (+0.1)	0 (−0.2)	0 (−0.2)	1 (+0.8)	0 (−0.3)	0 (−0.1)		2 (+0.9)	0 (−0.9)	
*Decreased*	1 (+0.7)	2 (−3.6)	1 (−0.1)	4 (+2.7)	4 (+2.7)	0 (−1.6)	0 (−0.9)		4 (−2.9)	8 (+2.9)	
*No change*	4 (−2.9)	**152 (+25.2)**	24 (−0.3)	21 (−7.4)	17 (−11.8)	30 (−5.7)	22 (+2.8)		163 (+8.3)	107 (−8.3)	
*Increased*	7 (+0.2)	106 (−19.8)	22 (−2.1)	35 (+6.8)	37 (+8.4)	42 (+6.6)	19 (−0.1)		148 (−5.5)	120 (+5.5)	
*Much increased*	3 (+2.0)	16 (−1.8)	6 (+2.8)	2 (−2.0)	4 (−0.1)	6 (+1.0)	1 (−1.7)		21 (−0.8)	17 (+0.8)	

The differences between obtained values and that expected with the Chi-square test of independence are represented in brackets. Data in bold are significantly overrepresented in comparison to predicted values (*p* < 0.01); ^1^ Chi-square test.

**Table 5 ijerph-17-06653-t005:** Relationship of mental severity and demographic factors.

Variables	Age Group (*n* = 590)								Educational Level (*n* = 590)		
	Below 18 (*n* = 15)	18–25 (*n* = 277)	26–35 (*n* = 53)	36–45 (*n* = 62)	46–55 (*n* = 63)	56–65 (*n* = 78)	66 or Above (*n* = 42)	*p*-Value ^1^	Post-Secondary or Below (*n* = 338)	Degree or Above (*n* = 252)	*p*-Value ^1^
Increased stress due to staying at home								0.000			0.006
*Not at all*	6 (+0.8)	52 (−44.2)	20 (+1.6)	27 (+5.5)	23 (+1.1)	**50 (+22.9)**	**27 (+12.4)**		108 (−9.4)	97 (+9.4)	
*Several days*	5 (+1.4)	63 (−3.2)	12 (−0.7)	17 (+2.2)	18 (+2.9)	17 (−1.6)	9 (−1.0)		70 (−10.8)	**71 (+10.8)**	
*More than half the days*	2 (−1.7)	**88 (+20.4)**	15 (+2.1)	13 (−2.1)	13 (−2.4)	10 (−9.0)	3 (−7.3)		92 (+9.5)	52 (−9.5)	
*Nearly every day*	2 (−0.5)	**74 (+27.1)**	6 (−3.0)	5 (−5.5)	9 (−1.7)	1 (−12.2)	3 (−4.1)		**68 (+10.7)**	32 (−10.7)	
Depressive severity								0.000			0.000
*Minimal or none*	9 (+1.8)	75 (−58.3)	28 (+2.5)	**41 (+11.2)**	40 (+9.7)	**61 (+23.5)**	30 (+9.8)		136 (−26.7)	**148 (+26.7)**	
*Mild*	2 (−1.3)	71 (+9.5)	15 (+3.2)	14 (+0.2)	12 (−2.0)	9 (−8.3)	8 (−1.3)		78 (+3.0)	53 (−3.0)	
*Moderate*	0 (−2.2)	**61 (+21.1)**	5 (−2.6)	3 (−5.9)	7 (−2.1)	6 (−5.2)	3 (−3.1)		**65 (+16.3)**	20 (−16.3)	
*Moderately severe*	2 (+0.6)	**45 (+18.2)**	3 (−2.1)	3 (−3.0)	3 (−3.1)	1 (−6.5)	0 (−4.1)		38 (+5.3)	19 (−5.3)	
*Severe*	2 (+1.2)	25 (+9.5)	2 (−1.0)	1 (−2.5)	1 (−2.5)	1 (−3.4)	1 (−1.3)		21 (+2.1)	12 (−2.1)	

The differences between obtained values and that expected with the Chi-square test of independence are represented in brackets. Data in bold are significantly overrepresented in comparison to predicted values (*p* < 0.01); ^1^ Chi-square test.

**Table 6 ijerph-17-06653-t006:** Pearson correlations between changes in physical and social health and mental health status.

Outcomes	(1)	(2)	(3)
Score of physical health (1)			
Score of social contact (2)	0.146 **		
Score of social support (3)	0.194 **	0.220 **	
Score of PHQ-9 (4)	−0.097 *	0.211 **	0.016

Note. * *p* < 0.05, ** *p* < 0.001 (two-sided *p*-values); (1): Score of physical health; (2): Score of social contact; (3): Score of social support.

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
