# Peer review of "Relationships between Physical and Social Behavioural Changes and the Mental Status of Homebound Residents in Hong Kong during the COVID-19 Pandemic"

_ijerph, 2020, doi:10.3390/ijerph17186653_

Round 1
Reviewer 1 Report
The authors did a wonderful job addressing all of my comments and recommend for publication as is.Author Response
Response to Reviewer 1 Comments
Point 1: The authors did a wonderful job addressing all of my comments and recommend for publication as is
Response 1: We are grateful for and appreciative of your expert comments.

Reviewer 2 Report
This study analyzes the effects of COVID-19 based on empirical data. The following points need to be considered.
First, a theoretical discussion related to this paper is needed. Despite the extensive theoretical discussions related to risk perception and depression, this study has no discussion of existing literature. A systematic theoretical summary, e.g., Paul Slovic's research works on the risk measurement paradigm, is needed.
Second, in this study, demographic variables such as age, sex, and job play the role of independent variables. Stigma, knowledge, and trust are frequently mentioned as independent variables affecting risk perception. It is necessary to discuss this.
Third, it is necessary to compare the analysis results with other data comparatively. Since there are many secondary data related to COVID-19, it is recommended to compare these data with the authors' data.
Author Response
Response to Reviewer 2 Comments
Point 1: First, a theoretical discussion related to this paper is needed. Despite the extensive theoretical discussions related to risk perception and depression, this study has no discussion of existing literature. A systematic theoretical summary, e.g., Paul Slovic's research works on the risk measurement paradigm, is needed.
Response 1: We have now incorporated risk perception in the discussion. Please see lines 264 to 274 for the revision.
Point 2: Second, in this study, demographic variables such as age, sex, and job play the role of independent variables. Stigma, knowledge, and trust are frequently mentioned as independent variables affecting risk perception. It is necessary to discuss this.
Response 2: We have now included knowledge, trust and virus information from friends and family as predictors of risk perception. Please see lines 267 to 268 for the revision.
Point 3: Third, it is necessary to compare the analysis results with other data comparatively. Since there are many secondary data related to COVID-19, it is recommended to compare these data with the authors' data.
Response 3: We have now included comparison of data regarding the prevalence of depressive symptoms in the discussion. Please see lines 261 to 264 for the revision.

Reviewer 3 Report
The authors revised the manuscript and improved the result section considerable. Very informative tables. Thanks
Regarding your results, recent studies (e.g. https://www.ncbi.nlm.nih.gov/pmc/articles/PMC7061893/ https://www.mdpi.com/1660-4601/17/9/3165 or https://royalsocietypublishing.org/doi/10.1098/rsos.200644 )
as well as
https://www.cambridge.org/core/journals/psychological-medicine/article/emotional-distress-in-young-adults-during-the-covid19-pandemic-evidence-of-risk-and-resilience-from-a-longitudinal-cohort-study/BD42C8C4EDFEEC6255554B195EA4ADED
also found increased stress levels in younger (18-25 but not below 18 and not for middle age people - allowing you to generalise beyond Hong Kong
Major issue:
which ethic committee approved the study? The information is still missing. If no ethical approval is required, please state so. But you asked about sensitive information (health status) and depressive symptoms. As you noted quite some scored over the cut-off point and should seek help for depression. Did you provide at the end some information where your participants can receive (online) help?
The PSA COVID-19 studies, do provide that, see https://psysciacc.org/studies/psacr-1-2-3/
Minor issues:
due to multiple tests performed, using p < .05 as significance level is problematic. I advise to use a stricter criterion, e.g. p < .01
it does not change the conclusions
p values on page 5 (chi square test) should be reported exact, not as p < .05 (see APA rules)
line 167 p < 0.001 - the first 0 is in italic, which it shouldn't
line 135: wrong reference, should be #24 not #23
line 171, 192, 193, p = .000 should be p < .001
line 197-199 - strictly speaking this sentence belongs in discussion
line 207 sentence is ambiguous, is the 18 to 25 age group showing more depressive symptoms or had more time to relax?
line 242/243 transmission of the coronavirus. "the" missing
line 242, 246, 285 - delete "in order"
line 253, comma before respectively
line 308 caused - d missing
Author Response
Response to Reviewer 3 Comments
Point 1: Regarding your results, recent studies (e.g. https://www.ncbi.nlm.nih.gov/pmc/articles/PMC7061893/ https://www.mdpi.com/1660-4601/17/9/3165 or https://royalsocietypublishing.org/doi/10.1098/rsos.200644 )
as well as
https://www.cambridge.org/core/journals/psychological-medicine/article/emotional-distress-in-young-adults-during-the-covid19-pandemic-evidence-of-risk-and-resilience-from-a-longitudinal-cohort-study/BD42C8C4EDFEEC6255554B195EA4ADED
also found increased stress levels in younger (18-25 but not below 18 and not for middle age people - allowing you to generalise beyond Hong Kong
Response 1: We are grateful for and appreciative of your literatures.
Point 2: which ethic committee approved the study? The information is still missing. If no ethical approval is required, please state so. But you asked about sensitive information (health status) and depressive symptoms. As you noted quite some scored over the cut-off point and should seek help for depression. Did you provide at the end some information where your participants can receive (online) help?
The PSA COVID-19 studies, do provide that, see https://psysciacc.org/studies/psacr-1-2-3/
Response 2: At our institution, ethical approval is required for research involving human subjects. The research team submitted the Human Subjects Ethics Review Checklist to the Research Committee (RC) of our College. No ethical approval was required as the study did not involve subjects who were unable to give informed consent, and there were no financial inducements being offered to the subjects.
The subjects were recruited from among staff and students of our institution, and members of the collaborating Hong Kong College of Community Health Practitioners. During the past months, information and advice concerning COVID-19 and related health issues have been disseminated to them by both organisations, including mental wellness.
Point 3: due to multiple tests performed, using p < .05 as significance level is problematic. I advise to use a stricter criterion, e.g. p < .01
it does not change the conclusions
Response 3: We have now reported the significant findings with p < 0.01. Please see lines 132, 149 to 150 for the revision.
Point 4: p values on page 5 (chi square test) should be reported exact, not as p < .05 (see APA rules)
Response 4: The p values on page 5 (chi square test) are revised. Please see lines 149 to 150, 171 to 173, 192 to 193 for the revision.
Point 5: line 167 p < 0.001 - the first 0 is in italic, which it shouldn't
Response 5: The sentence is revised in line 167.
Point 6: line 135: wrong reference, should be #24 not #23
Response 6: The reference #23 is correct in line 135.
Point 7: line 171, 192, 193, p = .000 should be p < .001
Response 7: The p values are revised in line 171, 192, 193.
Point 8: line 197-199 - strictly speaking this sentence belongs in discussion
Response 8: The sentence is move to line 306 to 308.
Point 9: line 207 sentence is ambiguous, is the 18 to 25 age group showing more depressive symptoms or had more time to relax?
Response 9: The sentence is revised as “The 18 to 25 age group scored significantly higher in the total score of PHQ-9 than the below-18 age group.” in line 205.
Point 10: line 242/243 transmission of the coronavirus. "the" missing
Response 10: “the” is added in line 243.
Point 11: line 242, 246, 285 - delete "in order"
Response 11: “in order” have been deleted in line 242, 246, 285.
Point 12: line 253, comma before respectively
Response 12: Comma is added in line 253.
Point 13: line 308 caused - d missing
Response 13: “d” is added in line 308.

This manuscript is a resubmission of an earlier submission. The following is a list of the peer review reports and author responses from that submission.
Round 1
Reviewer 1 Report
It is considered to be a very meaningful paper in that it is an empirical study on COVID-19. The following points need to be considered.
First, it is necessary to clarify the theoretical background of this study. Since many studies have been conducted, whether it is risk perception or depression, it is necessary to summarize the prior studies and then draw hypotheses to be demonstrated in the studies.
Second, this study adopted simple frequency analysis and cross-analysis, which are very descriptive in analysis methods. Additional analysis considering causal factors is required. In particular, the relationship between cause and effect needs to be verified through regression analysis.
Third, in addition to age and education level, income or social class are important variables, and the impact of these variables should be analyzed.
Fourth, it is necessary to clearly present the limitations of this study.
Reviewer 2 Report
Dear Authors,
Great work getting this much needed study rapidly into the field and disseminated. Some minor points of clarification are needed to assist non-scientific readers in interpretation:
1) please describe the mental health results as "symptoms" because the excellent measures used is a tool that only licensed professionals can utilize to assist in a diagnosis
2) please enhance the study section with all inclusion criteria, recruitment efforts, and online survey instrument process (for example, was it on social media and anyone could take it? was it emailed to certain people? etc.)
3) please include a few sentences within the discussion section that acknowledges the larger proportion of 18-25 year old within the study and how that could have influenced results
4) also include point three within the limitation section
5) please acknowledge the fact that these results are limited to those who took the survey and cannot be directly extrapolated to all in Hong Kong or in other areas nor for other time periods due to cross-sectionality.
Thank you, again, for this much needed study!